# Coordinated Interpersonal Behaviour in Collective Dance Improvisation: The Aesthetics of Kinaesthetic Togetherness

**DOI:** 10.3390/bs8020023

**Published:** 2018-02-09

**Authors:** Tommi Himberg, Julien Laroche, Romain Bigé, Megan Buchkowski, Asaf Bachrach

**Affiliations:** 1Department of Neuroscience and Biomedical Engineering, Aalto University, 02150 Espoo, Finland; tommi.himberg@aalto.fi (T.H.); megan.e.buchkowski@jyu.fi (M.B.); 2ICI-Project, Labex Arts H2H, Université Paris 8, 93526 Saint-Denis, France; lajulienroche@gmail.com (J.L.); romain.bige@gmail.com (R.B.); 3Akoustic Arts, 157 Boulevard MacDonald, 75019 Paris, France; 4EA 7410 SACRe, Université Paris Sciences et Lettres/École normale supérieure, 75230 Paris, France; 5Department of Music, Mind and Technology, University of Jyväskylä, 40014 Jyväskylän yliopisto, Finland; 6UMR 7023 CNRS/Université Paris 8, 75017 Paris, France

**Keywords:** improvisation, kinaesthetics, togetherness, coordination, interpersonal behaviours, agency, mirroring, rhythm, movement analysis, embodiment, enactivism

## Abstract

Collective dance improvisation (e.g., traditional and social dancing, contact improvisation) is a participatory, relational and embodied art form which eschews standard concepts in aesthetics. We present our ongoing research into the mechanisms underlying the lived experience of “togetherness” associated with such practices. Togetherness in collective dance improvisation is kinaesthetic (based on movement and its perception), and so can be simultaneously addressed from the perspective of the performers and the spectators, and be measured. We utilise these multiple levels of description: the first-person, phenomenological level of personal experiences, the third-person description of brain and body activity, and the level of interpersonal dynamics. Here, we describe two of our protocols: a four-person mirror game and a ‘rhythm battle’ dance improvisation score. Using an interpersonal closeness measure after the practice, we correlate subjective sense of individual/group connectedness and observed levels of in-group temporal synchronization. We propose that kinaesthetic togetherness, or interpersonal resonance, is integral to the aesthetic pleasure of the participants and spectators, and that embodied feeling of togetherness might play a role more generally in aesthetic experience in the performing arts.

## 1. Framing Togetherness

Moving together with other people brings us intrinsic pleasure. People move together while playing music, sharing the rhythm of their breaths and bodies together with the sounds [1]. Also, in spite of the agonistic aspect (competition) of sports and games, players are tuning into each other (cooperation, [2]). In dance contexts, this tuning-in aspect of moving together with one or more people is an essential part of the practice, often referred to as qualities of “listening” or “awareness” of self and others [3].

In this paper, we envisage a category of dance participatory events that we have come to designate under the umbrella of collective dance improvisations (CDI). A minimal definition of a CDI is that it consists of a collective event involving at least two people where the activity of the dancers is based on and regulated by the different qualities they experience in moving together. Examples of CDI can be found in forms of traditional round dancing such as ballroom dancing, and in forms of contemporary dance partnering such as Contact Improvisation [4]. However, we mainly envisage CDI in more experimental settings, and invent our own tools to design dancing scores and borrowing others from contemporary and post-modern improvisers [5], such as Steve Paxton, Lisa Nelson, Barbara Dilley, Julyen Hamilton, whom most of the dancers in our group trained with.

In CDI, moving together is not merely a by-product of the activity, but can be its very aim. We consider this aspect in its wider definition, where moving together is not merely the quality of similarity or synchronicity of the movements, but intrinsically related to the *feeling of togetherness* that arises in interaction. We see this “aesthetic experience of togetherness” stemming from the improvisers’ regulating their movements to favour their own and collective aesthetic pleasure of moving together. We share the view of Barbara Montero in that aesthetics are not only concerned with the aesthetic properties of a product, but can also concern activities leading to these products, and indeed we also see “proprioception as an aesthetic sense” [6].

Thus our approach of collective improvisation is an attempt to counteract the usual view that aesthetic study focuses on external properties of objects (be they dancers moving on a stage), and on the experience of a single person. It is true that Kant already argued that our judgements of taste “spill over” our subjective apprehension of the objects we consider, but this desire to share the experience with others remains highly individual [7], and Kantians hardly reflect on the actual activity of sharing. Aestheticians also rarely consider this: the paradigmatic concepts of beauty, harmony, elegance—the usual questions of aesthetics—are rarely if ever envisaged as collective experiences.

New forms of relational art [8] have shifted the attention of contemporary aestheticians towards the activities of these audiences as members of wider collectives. Concerning the collective aspect of aesthetic experience, a number of recent scientific studies on music and film reception have explored the effect of collective spectating on individual emotional or aesthetic experience [9]. Other studies have collected data from multiple spectators attending the same performance but did not frame their study in terms of collective experience [10,11,12].

Because contemporary artists are more and more involved in shaping relationships within the audience, it has become an important matter for philosophers of art to design relational aesthetics. Our aesthetics of togetherness is thus also a way to contribute to these relational aesthetics, by asking what is the specific joy of being and doing things together.

Collective dance improvisation allows us to answer this question by narrowing its frame around the particular experience of moving and being moved together, around the kinaesthetics of togetherness, that is: the specific motor aspects of the experience of togetherness. This active, kinaesthetic togetherness is more than just the joining of individual feelings or states of mind about a shared situation, it is the joining of actions, gestures or movements in a shared situation. In the experiments we have conducted, not so surprisingly, the kinaesthetic value most favoured by co-movers had to do with smoothness of co-gestures, the absence of jitter in co-gestures being indicative on a motor level of the tuning of individual decisions to each other [13].

Even though we describe the dancers’ experience as an aesthetic experience, it does not mean that we think that the “maker’s” point of view is the paradigm for understanding the experience of the “receiver”. Instead, we propose that the experience of dancing, and specifically of dancing together with other people (like the experience of making music together, building together, and playing a game together), rests on specific perceptual and dynamical mechanisms. Our social approach would aid in understanding the aesthetics of new forms of relational art, and could also shed new light onto the aesthetic experiences of witnessing more traditional works of art.

We will thus consider dance-making and dancing (specifically focusing on forms of “free improvisation” or “experimental improvisation” based on contemporary dances such as Contact Improvisation, Instantaneous Composition etc.) not only as the art of making dances to be seen by audience (cf. [12] for a recent study on the aesthetic experience by spectators of movement synchrony in contemporary dance improvisation), but more specifically, as the art of framing the activity of a collective of movers in such way that they are able to experience (and hopefully enjoy) their own collective agency.

## 2. The Aesthetics of Agency

This point of view on dance-making as an art of framing collective agency makes it a special case of what American philosopher Thi N’Guyen has called the “arts of agency”, that is: the arts of framing an experience for agents to enjoy their own aptitude to respond to a given situation [14].

To experience one’s own agency in a given situation does not necessarily require specifically designed conditions: it can happen spontaneously when doing home improvements with randomly selected materials, or when improvising a dinner “from scratch” because friends invited themselves over at the last minute. In those cases, we face unexpected obstacles, and have to make do with what is at hand. However, although this can happen spontaneously, some contexts aim at fostering this experience and some people have dedicated their works to favor those contexts.

Let us call these contexts “games” and those people “game-makers”. Games are defined by Bernard Suits as “the voluntary attempt to overcome unnecessary obstacles” [15]. The first specification that we gain from this definition is that we can differentiate games from other goal-oriented activities. Gamers pursue a goal (sometimes winning the game, sometimes supporting the prolongation of the game; this depends on the game itself), but they do so under special conditions where not every tactic can be used. By contrast, most technical activities will use any means necessary to complete a task, for example, in order to drive a nail to the wall, it does not matter if we use a hammer or a shoe, as long as the nail ends up in the wall. To win a marathon, on the contrary, it does matter whether we run or take a cab to the finish line. In a game, the means count at least as much as the goal itself, if not more. In sports, we can say that someone has played “a good game” even if they lost; in chess, we can say that someone has made a “beautiful move” even if they were ultimately defeated.

These expressions help us move towards the aesthetic understanding of games. In a game, at least as an audience, we ascribe paradigmatically aesthetic features to some gestures or attitudes not because of the purpose they serve, but because of a specific fitness or harmony that we ascribe between the apparent resources of the player, and the difficulty of the situation. However, this is not enough: this does not explain why people play games, not only in front of an audience, but also by themselves. In some games, “the experience of game-playing is more than simply spectating one’s own externally available actions from the closest possible seat in the house. There are distinctive experiences only available to the causally active game player. These are the experiences of acting, deciding, and solving—of not only appreciating the movement or solution, but originating it, and originating it in response to recalcitrant opposition.” [14].

And these experiences of acting, deciding, solving, are judged internally, at least from time to time, within the same aesthetic categories (elegant, graceful, clumsy, etc.) that are usually used to describe the experience of artworks. This is one of the reasons why aestheticians should look more closely at the aesthetics of games.

## 3. The Kinaesthetics of Togetherness

Our specific interest lies, not in this possible extension of the aesthetics of agency to the activity of witnessing art, but in a further specification of agency. We want to look at the phenomenon of collective agency in an art-making context—that of dancing together.

In dancing together, many phases are encountered, including phases of dis-adaptation, awkwardness, negotiation with the others, calls for recognitions, solitary wandering, etc. Among those phases, we investigate one phase that is specifically sought after by the dancers: dancers having the experience of togetherness. This experience of togetherness is separate from just moving in synchrony. For example, moving together in a highly coordinated fashion without a sense of togetherness can be experienced daily on a busy sidewalk or a metro station. In dance, moments of synchrony without the experience of togetherness are not necessarily failures to achieve togetherness, but can have other goals and hold an aesthetic value of their own (which we will not explore further here).

On a physiological level, experiencing movement rests on two distinct functions. The first function corresponds to the *kinaesthetic sense*, and by that we understand all the sensory systems involved in manifesting ego-locomotion, on both an internal level (muscles, joints, viscera, vestibular system, etc.) and an external level (variations of the field of vision, variations of the acoustic shape of space, etc.) [16]. The second function corresponds to the *detection of movements*, and specifically in our case of dance-partners, in the shared milieu of action, which mainly involves distal senses (such as the visual and auditory systems) although being in contact can provide also proximal feedback of the others’ movements.

Experiencing togetherness in dance involves the crossing of those two functions: kinaesthesis and movement detection. This crossing is transmodal or amodal, which means that it crosses or precedes sensory boundaries (the togetherness of my movements and your movements can be sensed between my visual apprehension of my ego-locomotion and my auditory apprehension of your locomotion). In the case of collective improvisations involving up to thirty people dancing together, it is essential to recognize this transmodality of experience of self and other.

On a phenomenological level, the aesthetic experience of togetherness rests on the emergence of a specific sense of agency. This specific sense of agency does not consist only of the connection between individual’s abilities and the demands of the situation, as was the case in the games previously described. To this self-centered satisfaction, a second degree of connection is added, that articulates not only “I” and the situation, but “we” and the situation.

In a collective dance improvisation, it sometimes happens that our agency is distributed among us: instead of the experience being lived dually as an addition of subjectivities, it is lived as a joint movement where the subject is a collective rather than a sum of individuals. Verbal accounts of such experiences thus tend to use the “we” pronoun (“when *we* were doing this...”) rather than the “I” and “you” pronouns that could describe the situation as well (“when *I* was doing this, *you* were doing this too...”). We propose to consider that this is not a “shortcut” or a manner of speaking, but a way of pointing to the collective aspect of the lived experience.

The displacement of subjectivity from the “I” to the “we” point of view indicates another important phenomenological feature of togetherness: as we move together, the boundaries between moving and being moved blur, and we do not know with certainty whether we initiated a movement or inserted ourselves in a pre-existing wave. The experience of moving together thus brings the subjects to a non-dualistic apprehension of themselves, where they seem to accept the compatibility between self-centered and we-centered perspectives. This does not mean that the choice dimension disappears, but simply that the entity considered to be making the choices is flickering between I and we: as we move together, I shift from decision-making to surrendering to our movements [17].

First-person testimonies of the experience thus do not show a loss of the individual perspective, but rather seem to indicate an augmented, or enlarged individual perspective, a we-centered perspective (first person plural). This paradoxically tends to reinforce the subjective view, for example, the moments that subjects describe as moments of togetherness seem to be more precise in their recall. It has been noted that interaction creates a self-other interdependency rather than dilution of self-agency; the two are not mutually exclusive but rather, the we-agency builds on strong self-agency [18,19].

Why are movers not experiencing moving and being moved (by each other) as two distinct experiences? It might simply be because their movements are of a pre-subjective or a-subjective nature. It is possible that the shared movements precede their individuation in each mover. Speaking of pre- or a-subjectivity greatly simplifies our notion of togetherness: it no longer needs to be a matter of face-to-face or individual-to-individual interaction between separated subjectivities (as is classically understood with the concept of inter-subjectivity); our togetherness relies on dimensions of our experience that bypass and precede individuations and remain latent in our individuated lives [20].

How can we understand and describe this pre-subjectivity? One option is to use the concept of “vitality affects” [21]. Vitality affects are what is shared between infant and mother in their “first relationship”: like our kinaesthetic togetherness, vitality affects are sensed beyond, or rather before from the infant’s point of view, the division and formation of the senses into systems. They consist, more specifically, of dynamic impressions that have specific “activating contours” common to internal states (such as heartbeat, breathing), mental states (such as the speed of thought, a clarity or confusion of mind) and externalizing behaviors (muscle tone, gestures, sounds). In the domain of movement analysis, vitality affects can be understood as (and, in effect, have been inspired by) the subjective, or felt, counterpart of Laban’s Effort system [22], or Kestenberg’s Tension Flow attributes [23]. This amodality permits the tuning of the mother’s and infant’s affective states beyond recognizable emotions in “vitality forms” such as rushing, exploding, crescendo, decrescendo, etc. [24].

In what sense are vitality affects pre-subjective? It is the activity of tuning that explains the pre-subjectivity of the vitality affects: vitality affects are not something that infant and mother would have in the back of their minds and would then try to impose to each other; vitality affects are the very dynamic of sharing that occurs between infant and mother. In other terms, the vitality affects do not exist, at least for the infant, apart from their negotiation with the mother to tune with her.

Whether those vitality affects persist with adults as remainders of their ontogenetic origin or not is not a question we intend to solve here (for a convincing correlation between Stern’s theory and Simondon’s theory, see [25]). We simply want to point out, with this description, the existence of certain pre-subjective affective dynamics that do not rely on internal, subjective states, but on shared and common negotiations.

Our understanding of collective dance improvisation, beyond being simply the sum of individual movers, relies on a similar idea of the aesthetic experience of togetherness. This being not only as a singular, individual strive for fitting in the group, but as a collective endeavor to move together. This collective endeavor, much like the “beautiful moves” of the chess player and the “elegant gesture” of a tennis player, does not specifically rest on the desire to reach a common exterior goal, although having a goal allows participants to strive for it together. The aesthetic experience of togetherness lies in the very tuning of our actions together, and in the meeting of our movements.

It is the activity of *tuning-with* that we seek to investigate experimentally, resting on a variety of “games” going from ecologically-reduced situations such as hand mirror games to fully developed dance improvisations in a studio space.

Our proposition underlines the collective dimension of both the experience of togetherness and of the process of “tuning with”. In a collective endeavor such as CDI, both this process and this phenomenon are non-dualistic in nature, at least in the sense that I am no longer in front of my partners, but with them. Taking them as objects of study is therefore at odds with the traditional frameworks of cognitive sciences, in which the dualism between “I” and the others is taken for granted. Indeed, in classical cognitivism, the focus has been put on the abstract processes that are at play in individual minds and that allow us to represent the world and act within it [26]. Consequently, in this view, social interactions have often been reduced to a mere succession of individual processes happening in each isolated mind; namely, “social cognition” processes that allow us to represent [27] or simulate [28] others’ mental states and to interact with them accordingly (for a critic, see [29]). Here, the dualistic separation between the “I” and the “others” is taken as a starting point to study interactions, while the collective dimension of the interaction process itself do not play a role in the constitution of their behaviors and experiences [30]. It is, therefore, difficult to articulate co-agentive phenomena in the context of a classical cognitivist framework. Furthermore, the traditional focus on abstract, computational processes prevents the investigation of the dimensions of experience that are related to feelings. To interrogate the lived experience of togetherness and its non-dualistic nature, and to experiment on the collective and shared dimension of the activity of “tuning-with”, our methodology requires a paradigmatic move that bends this traditional, still-dominant view on social interactions. This is why we endorse the current shift towards the enactive and dynamical approaches, as they provide frameworks that fit our concerns better. Indeed, as we shall see below, these approaches can better account for the interpersonal (or rather, “transindividual”, in Simondon’s terms) origins of some behaviors and experiences, and they do so by taking the role of the coupling between agents seriously.

## 4. An Enactive View of Togetherness: Bodying-Together

The enactive approach describes individuals (and living systems more generally) as autonomous beings. It means that they operate according to their own laws of organization. Their main principle of organization is autopoiesis [31]. This means that living systems are producing themselves and, by doing so, they define what they are; they affirm their own identity. By the same token, they also define, from their own point of view, what counts for them as an environment (the notion of environment thus contrasts with the notion of “objective space”, which a behaviorist observer believes they can define to describe a person’s actions). To keep self-producing, living systems have to interact with this environment: their autonomy depends on the viability of their relations between themselves and the environment. A constant movement of bodying is thus defining the existence of animate beings, through which they weave their existence to their associated milieu [25].

Structural embodiment (i.e., having a body that can sense and move) is the concrete realization of autopoiesis, as well as the enabling condition of interactions. Accordingly it is the enabling condition of the sustaining of autopoietic processes. Autonomous embodiment is thus what gives a perspective or point of view to a living system: it anchors the system to a reference point, defines what can be sensed and done, and defines a proper world to be experienced by the system. Since motility affects what is sensed, “bodying” is the source of what can be known and how it gets to be known. Here action and perception are not separate processes, but they co-exist in a continuous coupling. The active regulation of this sensorimotor coupling constitutes our experience and it does so by dynamically coupling our being with the world.

In our case, we are interested in interpersonal interaction. As we shall see, this particular type of interaction opens up a new phenomenal domain. In this case, indeed, there is a strong structural similarity between the embodiment of interacting partners: we can sense and act upon similar aspects of the world. In the context of an interpersonal interaction, what we do does not only change what we sense, but it also changes what the other senses, and therefore what they do and what we further sense in return: our sensorimotor couplings get coupled together [32]. As a result, part of our experiences are shared, since they are situated in this mutual coupling (my experience is constituted partially by what constitutes the other’s experience). In short, we co-constitute and incorporate each other’s experience [33].

If I influence the other by my actions, and if they change me as well, then I change myself by interacting with them. This makes the interaction process constantly move forward and our experiences and behaviors get caught in these continuous dynamics. The dynamics of the interaction process that we give rise to, constrains us and affects us mutually, and therefore by coupling our respective sensorimotor couplings, coordinates our behaviors and experiences to some extent. The potential for sharing personal experiences is therefore in the interaction process itself, thanks to the reciprocal affection that it allows for. In the context of collective dance improvisation, the very object of the interaction process seems to rely on this affective attunement of the dancers, whose ability to recognize togetherness is not separable from its affective/aesthetic dimension.

The coordinative effects of interaction have been observed experimentally in many contexts such as basic motor tasks (e.g., [34,35]), sports (e.g., [36,37,38]), music (e.g., [39]) conversations (e.g., [40,41,42]), and beyond (see [43,44]). It has been shown to manifest itself whether participants were aware or willing to coordinate or not, or even when they are instructed not to coordinate with each other [45]. However, this only means that a collective pattern can be observed from the outside.

Proponents of the enactive approach conducted a series of study that looked for the minimal conditions under which interpersonal coordination happens. In the principes experiment [46], participants moved an avatar in a minimalist virtual reality. Their sensorimotor coupling was reduced to its minimal structure thanks to a device that produced tactile stimulation on their finger whenever they were making an encounter with another agent in that virtual space. Three types of agents could be encountered: an immobile agent, another participant’s avatar, or a lure that was imitating the partner’s avatar at some distance. Participants had to click whenever they felt in presence of the other. They were only able to discriminate moving and non-moving agents, but not the partner from the lure. The study found that despite lack of discriminatory information participants spent more time interacting with the others’ avatar than with their lure. The explanation is that they were mutually affected by the situation and got attracted towards each other’s position. Here the interaction process coordinates individual behaviors beyond will or recognition; they found each other and stayed together before they knew it. The experimental design produced patterns from the third-person point of view, as well as a dynamical process that is proper to the interaction and which coordinates individual behavior in a pre-reflective manner. However, this was not enough to give rise to a specific first-person experience of togetherness.

It is important to note that the task was individual in nature: “recognizing the presence of the other”. Participants were not instructed to accomplish the task together. Froese et al. [47] changed the nature of the task by turning it into a team tournament. Participants had to find each other in the virtual reality as a team/duo. This time, participants sought to regulate the interaction process together. In this context, active co-regulation of the interaction process seemed to be necessary for the awareness of the other. Interestingly, these moments of awareness tended to occur at similar moments of the interaction for the two partners. In other words, under minimal conditions, the active co-regulation by the participants of an interaction process might give rise to specific dynamics that can be both observed and experienced.

## 5. Investigating Togetherness: The ICI Project

The interaction process gives rise to its own dynamics, which constitute a proper level of analysis. In continuity with the first part of the paper, we name this level the ’first person plural’ perspective. These dynamics constrain individual experiences (first person singular perspective) and behaviors, and therefore changes and coordinates them. This gives rise to patterns that can be measured from an observer point of view (third-person perspective). To experience togetherness from the first-person point of view, however, it is necessary to participate actively in the co-regulation of the dynamics of interaction.

We propose that the phenomenon of togetherness exists as the conjunction of these three perspectives (first singular, first plural and third person) and can be defined as a dynamic negotiation between two or more individuals experiencing their movements in a non-dualistic way, involving at the same time their individual points of view (I-centered) and a collective subject of movement (we-centered).

As a consequence, our project aims to address the embodied aesthetics of togetherness through data concerning the movements as they are deployed in space and time, and through first-person testimonies concerning the kinaesthetic and affective experience of the movers. Accordingly, we aim at a global methodology in which togetherness is measured and observed at these three levels of analysis.

The ICI project (from joint improvisation to interaction) is a Paris-based research group composed of cognitive neuroscientists, dancers and philosophers. The aim of our group is to develop, through joint improvisation, transdisciplinary research and intervention programs regarding social interaction. The first and crucial step of our work process consisted of a multi-directional immersion. Scientists and philosophers danced, dancers were brought into the language, tools and mind frames of experimental science and philosophy that assisted with articulating embodied practice and shared concepts between disciplines. The second step was the collective elaboration of short (up to one day) test-intervention protocols building on the shared understanding and common questions formulated in the first stage. In parallel to the scientific-artistic research, our interactions brought about a number of artistic performances, installations and dance intervention projects.

### ICI’s Ten Design Principles

Through the collaborative process we have identified ten ‘design features’ or principles that have guided the development of the different protocols or experimental scores (see Table 1). These principles are the practical implementation of the theoretical perspective described in the first section of the paper.

**(1)** The first principle, already alluded to before, is *design through experience*. All our protocols are the fruit of shared experiences and are often the surprising result of the juxtaposition of a scientific finding, question or method and an improvisational device. In other words, we have chosen to use specific protocols/scores to study togetherness if we, as researchers, have ourselves had experiences associated with togetherness while practising or elaborating the score.

**(2)** The second design feature is *open ended-ness*. We favor activities which are not organized around an end product or a target. This is quite different from the standard approach to the study of joint action in the scientific literature, where participants are given a specific task, such as synchronising repetitive movements together. This feature comes from our experience of dance improvisation. It seems to us (based on experience, not scientific quantification) that goal-lessness focuses the participant’s attention on the process itself and to the interaction (meta-attention) and so allows for a richer first person perspective and eventually for more significant learning.

**(3)** The third design feature is *poly-solution*: There is never only one way or best way to go about accomplishing the task. In fact, as in the case of the four-person mirror game below, there might actually be no way at all to accomplish it. This, we believe, is a critical feature that allows for choice-making and fosters improvisational set of mind (creative divergent thinking). A solution space where multiple options are all ‘as good’ is particularly interesting when considering collective improvisation, since it requires negotiation and listening at the group level (group coordination cannot be reduced to individuals making optimal choices). It is critical however to also limit the space of choices in order to make the group interaction possible/meaningful for the participants and for the (scientific) observers.

**(4)** The fourth design principle is *interactive generativity*: In a generative system, new outputs are produced via the (iterative) composition of basic elements [48]. We name “interactive generativity” the observation that through interaction, simple actions or gestures can serve to generate increasingly complex constructs (such as in language [49] or music [39]). Interactive generativity is, we believe, another hallmark of collective improvisation. Improvisation is often less about an individual coming up with a totally different, original or complex action, but rather about the collective composition of individual (‘simple’) propositions into a novel, complex and at the same time, appropriate whole.

**(5)** The fifth design principle is *always more than two*. Our protocols are designed so that they cannot be “solved” individually, but are collective in nature, and focus on the interaction process itself. Furthermore, in all our protocols we have sought to avoid dyadic situations, proposing to test and apprehend togetherness not as the sum of two individuals facing each other, but trying to multiply the points of view. Because of the collective (rather than dyadic) aspect of this principle, this strategy also serves to prevent, in ourselves, the tendency to come back to the “I” (subjective) and “they” (objective) points of view in favor of a we-inclusive perspective (collective). Indeed, the complexity of non-dyadic assemblies makes the regulation of the collective situation less trivial [50]. It therefore provides a better test-bed for the idea according to which the phenomenon of togetherness finds part of its roots in decentralized, non-individual processes.

**(6)** The sixth design principle is *measurability*. This design condition limits the poly-solution condition as it requires some form of repeatability of observations. Importantly the measurability principle regards both third-person observations (motion capture, physiological measures, etc.) and first-person data. The latter is particularly complicated and delicate. We aim at protocols that promote meta-attention to one’s own (or the group’s) state in real time (open-endedness is important here) and that allow for either real time or off line collection of these reports. We believe that our current protocols promote meta-attention to the interaction but we are still in the process of optimizing appropriate report collection (cf. [51] for relevant work with dance spectators, and [52] in the context of music improvisation).

**(7)** The seventh design principle is *experience-independence*. We aim at protocols that do not require experience in dance or improvisation so that novices (and eventually younger population and neuro-diverse individuals) can take part. At the same time, we want these protocols to provide a rich enough context for experienced improvisers to ‘play’ (feel that they are improvising and not executing a task). This is a great gate-keeper condition that prevents us from over-simplifying the protocols (in particular for the sake of measurability) and thus losing the artistic core.

**(8)** The eighth design feature is the *elicitation of shifts in the patterns of behaviors*. In non-linear dynamical systems, phase transition occur when a system abruptly bifurcates from one state to another. We believe that such bifurcations take place and are experienced during CDI. These are moments where the quality of the interaction/room suddenly changes in an athematical manner [53]. Identifying such moments (from first person singular, plural and third person perspectives) is of critical importance for the understanding of the affective dynamics of CDI and interactions more generally. In our protocols we try to create conditions that promote such shifts and facilitate their observation/measurement.

**(9)** The ninth principle concerns the *primacy of the kinaesthetic experience*. Our protocols minimize verbal exchanges or the use of visual support and highlight the experience of (shared) movement as the basis for (inter)action. Instead of relying on complicated, verbal instructions to give to the participants in the beginning of a “task” (see (2) above), our protocols rely on providing only starting points (and perhaps some pointers along the way, as in the “rhythm battle” protocol below). The collective improvisation then grows from the kinaesthetic interactions of the participants, and is fundamentally embodied in nature [54].

**(10)** The final but maybe the most important principle is *fun/pleasure*. As we mentioned above, the pleasure of moving together is essential for the aesthetic experience of togetherness. We aim at protocols that remain pleasurable and engaging despite some required constraints and limitations, to meet the other principles (especially (6) measurability and (7) experience-independence).

Next, we describe two of the protocols that have been developing and piloted by our group along the lines of the design principles describe above. While these two protocols are specifically targeted at producing measurable movement-based (principle 9) correlates of togetherness they are the product of first-person experience in joint improvisation (1). The pilot experiments reported here represent the first stage of the construction and validation of these protocols. We wanted to establish that these protocols are accessible and engaging (7, 10), that they let emerge complex interactive patterns from simple instructions (4) and that they bring about variability of solutions (3) while remaining tractable and measurable (6).

In ongoing work and future steps of this research program we combine these ‘third-person’ measures with online and offline first- and second-person accounts of togetherness.

## 6. Two Examples of ICI Protocols

### 6.1. Four-Person Mirror Game

The dynamics and kinematics of dyadic hand movement mirroring have been studied previously [13,55]. In these studies, the condition where participants share leadership has, somewhat surprisingly, produced the smoothest and most synchronised performances. Being attuned to each other seems to trump having well-defined roles in this game.

The emergence of jitterless, “coconfident” movements [55] indicates that participants are engaged in an intersubjective, de-individuated state. The mirror game thus holds much potential, both as a context for studying attunement and the emergent, intersubjective behaviours, and as an enactive measure of dyadic intersubjectivity, as suggested by Yun et al. [56]. We were interested in expanding the mirroring beyond the dyadic setting, in line with the general aims of our project discussed above, and seeing if analysing group mirror games preceding and following different types of improvisations, could shed light on to the intersubjective processes influenced by these “interventions”.

In our version of the four-person game, participants stand in a circle, arms extended towards the centre of the circle. Compared with the two-person game, the four-person game creates conflicts for the participants to solve: they need to decide whether to mirror or mimic lateral movements, and with whom to move, if conflicting movement impulses emerge. This makes the four-person setup good for studying group dynamics and implicit social biases and preferences, and thus a potential measure for the socio-affective effects of various interventions. From the perspective of the mover, the impossibility to mirror all three partners, and the consequent need to make choices, turns the four person game into what we consider to be a better model of ‘ecological’ joint improvisation practice (in comparison to the dyadic form).

The four-person mirror game ticks all the boxes of our list of design features. However, as the data reported here comes from a kinematic study conducted in a motion capture lab, measurability (feature 6, see Table 1) was emphasised in the design of the protocol, at some expense to the other features. For example, while even this version of the game was open-ended (2), in that participants were free to be creative with their hand movements, and that their only task was to create enjoyable movements together, their improvisations were constrained by the design. They were limited to just moving their hands, as they were asked to stand in a certain spot. Also, the improvisations they performed were relatively short. We envision that eventually the four-person mirror game could grow from this simple kinematic experiment into a properly open, creative improvisation, where perhaps the hand-movement mirroring provides just the starting point.

Changes in the game can either be observed in the kinematics and dynamics of the movements of participants (third person perspective), or are evident in the different experiences of participants (first person perspective). In two pilot studies, we measured participants’ movements using optical motion capture, and gauged their experiences through post-game interviews. The aim of these pilots was to observe how groups with different levels of improvisation expertise perform in the game, and also to measure how movement mirroring changes as a result of different social activities between the games. The motion capture system provides a 3D position map of the movements at high spatial and temporal resolution, helping us better characterise these behaviors, and to develop measures to be used in our more typical contexts: open, collective improvisation, in e.g., a dance studio or a classroom, measured using mobile sensors.

#### 6.1.1. Methods

*Participants:* Four experienced dancers took part in the first pilot. In the second pilot, we collected data from four groups of four participants with no to moderate prior experience in dance.

*Equipment and measures:* Participants’ body movements were tracked using optical motion capture (OptiTrack, 100 Hz sampling rate). Each participant wore a total of 12 reflective markers, placed on their head, upper body joints, and index fingers.

From motion capture data, we calculated two metrics. First, as a measure of overall movement coordination, the average quantity of motion (QoM) for each participant, or the average speed of all of their markers. This measure reflects how much they move and how their movements change over the course of the game. For a more detailed synchronisation analysis, we also calculated the cross-correlations of fingertip accelerations for each pair of participants. From the cross-correlation function, we extract the highest correlation (CC), and the time lag (LAG) at which it occurs, to get measures of how similar people’s movements are, and which of them leads and by how much [57].

*Protocol:* In the first stage of the protocol participants are familiarized with the two person mirror game (using extended arm and finger as in [13]). As generally practised in the literature, each participant has the opportunity to lead and follow before a shared leadership phase is introduced. In the second stage, the four-person mirror game, participants stand in a circle, one arm extended to the middle, and are asked to mirror each other’s movements without assigned roles (Figure 1C).

In the first pilot study, four experienced dancers played a 80-s mirror game before and after engaging in a synchronised group movement improvisation. The improvisation task started with the four participants holding a single A1 sheet of paper. They were asked to follow the paper with their eyes closed (and then with eyes open). In the second stage of the improvisation, the paper was dropped and participants were asked to continue moving as if still holding the paper. This improvisation score was chosen as we expected it to lead to a specific ‘joint’ behavioral solution to the mirroring paradox: keep the interpersonal space constant.

In the second pilot study, sixteen participants (9 female; mean age 28.2 years, with 3 ages missing; 50% had no musical or dance background, 12.5% had approximately a year of musical experience and 18.8% had more than 10 years of dance experience) played two-minute mirror games before and after an improvised game where they moved objects on the floor, taking turns. This improvisation was chosen as it had no relationship with the mirror game, and we expected it would only have a more general effect in the social bonding between the participants, but would not offer cues to solving the mirroring paradox.

Participants were debriefed in a free interview after the protocol. The interviews in the second pilot were recorded on video.

#### 6.1.2. Results and Discussion

*Pilot 1*: Figure 1A shows the QoMs for each participant, in the game preceding (top) and following (bottom) the improvisation. Comparing the two, the first game seemed to produce a less coordinated performance, as at any time, the four participants can move at very different speeds. Participants are not coordinated very well, and larger movements by one participant are followed by the others often with considerable lags.

Looking at the pairwise cross-correlations (Figure 1B, top), we can similarly see that maximum correlations in each pair of participants are quite low (peaks in the curves), and the pairwise peaks are at very different lags (x-axis values), indicating that the movements are not very similar, and there seems to be no group-level coherence.

In both metrics, there is a clear change between the first and second game. In the second game, the QoM profiles of participants are very similar. Even the large movements towards the end of the game, while being about twice the magnitude to the peaks in the first game, are very coordinated. Looking at the pairwise CC’s (Figure 1B bottom), the peaks are much higher, indicating better synchronisation, and also better aligned with each other, indicating that in addition to individual pairs of participants being synchronised to each other, there is also a group-level coherence. There is also an interesting pattern in these pair-wise correlations and their lags: the peaks with smallest lags (closest to zero, either negative or positive), occur in pairs that stand next to each other. Lags for pairs standing opposite each other are larger. This could indicate that physical proximity in the circle is one of the factors that affects the choices people make in solving the mirroring paradox. However, this might be just idiomatic to the solution in this game, as it was so strongly based on maintaining interpersonal distances.

*Pilot 2*: Overall, the four groups had different changes from pre- to post-improv game. Where one group clearly improved their coordination and synchronicity, two other groups showed virtually no change in these metrics. One group even got less coordinated in the post-improv game compared to the pre-improv game. Figure 1D shows all the pairwise QoM correlations, and the variability in this data, which clearly indicates that the slight trend for better coordination overall is not statistically significant. This is largely because in this type of statistical analysis we lose a lot of information and degrees of freedom, as each pair’s performance together during the two-minute game is simplified to just one value in each of these analyses.

Thus we need to look into the performances in a different way, and start from qualitative observations and what participants told about their strategies in the post-experiment interviews.

Generally, the groups in the second pilot seemed to have a very different strategy than the expert dancer group in the first pilot. Perhaps surprisingly, there were many commonalities in strategies between the groups in pilot 2. Instead of aiming to dissolve leadership in favour of co-confident movement and mutual following (as happened in the post-intervention game of the first pilot), participants in the second pilot shared leadership mostly by taking short turns in leading the game.

Experienced improvisers are familiar with the de-individuated state of dissolved leadership, and blending of agencies and perceptions of one’s own and partners’ movements. They can therefore collectively seek the state where movement “just emerges”. In contrast, in the post-experiment interview several participants in pilot 2 stated, “for a fact”, that there cannot be movement unless someone takes the lead and creates it. Perhaps the groups where people had little to no experience in improvisation practices were unable to find or even search for the solution where all mutually follow? Situational factors might also have contributed to this difference between pilots. The second pilot was not embedded into a dance class, but was more clearly a movement experiment. Thus participants might have felt more pressure to produce movement “at any cost” from the very beginning of the game.

Interestingly, it is also possible that the intervention (between the two runs) influenced their strategy. Whereas the paper-following in the first pilot primed the group for continuous synchronized movement, the object game intervention in the second pilot, a turn-taking exercise, might have primed the strategy of leading in turns. This turn-taking dynamic opens up interesting parallels with conversation analysis. Participants took turns in proposing movements and gestures for others to follow, and responded to proposals made by others. This very dynamic is currently under investigation in conversation analysis [58]. These studies look at e.g., how in dyadic decision-making, proposals are accepted, ignored or rejected, with each fate having a different effect on the interaction dynamics, such as body movement coordination and the conversants’ gaze behaviour.

To explore these proposals and responses to them, we annotated the video recordings of the games. The aim was to find the most salient cases of proposals and responses. As these proposal-response sequences are purely movement-based, we hope to be able to automatise this process in the future. We annotated accepted and ignored proposals, which participant made them, and which participant first responded to each proposal at what time lag.

Qualitatively, we identified different interaction patterns as response to proposals. For accepted proposals, it was notable that the lag between the proposal and responses to it varied widely, from immediate to multiple seconds.

Finally, we spotted multiple occurrences of the quartets splitting into two dyads. This could happen either by the two dyads having two different movement patterns, or more commonly so that all four had the same basic, repeating pattern, but with a time lag between the dyads. In contrast with the finding in pilot 1, where participants synchronised best with partners next to them, here the splitting often occurred so that participants facing each other formed the pairs.

Could the number of initiatives one makes in the group be linked to personality? To answer this we correlated the amounts of proposals, as well as the times a participant was the first to follow a proposal, with Big5 personality dimensions [59] and the perspective-taking subscale of the Interpersonal Reactivity Index (IRI, [60]). We did not find links between the Big5 and either proposals or first follows, but did find a positive association between the PT/IRI and being the first to follow other’s proposals (F=8.27,p<0.02).

The post-experiment questionnaire and debrief support the observations. Using the Inclusion of Other in Self Scale (IOS) [61], we asked how close the participants felt to their group members. In general, there was no strong relationships. Two slight trends emerged: Person 1 tended to feel closest to Person 3 (the person standing across from Person 1) in three out of the four groups, and Person 4 tended to feel closest to Person 2 (across from Person 4) in three out of the four groups. This suggests that people who were across from each other in the quartet tended to feel closer, although it was unidirectional in both cases.

In the debrief questionnaire and discussion, a few trends arose. The first was the necessity to have a leader in the shared leadership conditions. All groups strongly expressed that the game could not be played unless there was a leader “selected silently”. Simply, having a designated leader was much easier for the game to be played. Participants in pilot 2 appeared to have a difficult time understanding the possibility of shared leadership, and thus never achieved it. This challenge can be explored in future studies.

### 6.2. The Rhythm Battle

The aim of this protocol is to simultaneously test entrainment and resisting entrainment in a large group of participants. A large (more than 10) group of participants is divided into two teams who are then trying to maximise entrainment and tempo stability within the team, while minimising between-teams entrainment.

This protocol was inspired by a study on *Congado* [62]. Congado is a Brazilian tradition where marching bands parade around the city, playing and dancing. When two bands meet each other, they attempt to maintain their own tempo and try not to be influenced by the other group. Losing one’s own tempo is to lose something of one’s identity. Lucas et al. showed that entrainment is difficult to resist, as it is a strong, automatic tendency. Groups would often entrain, or adapt to each other’s tempi. Entrainment in a more complex, n:m pattern (one group has n beats in the time the other has m), was also observed.

Rhythm battle, that can be carried out without drums or other instruments, combines entertainment with resisting it into the same task, and therefore has especial potential for studying in-group–out-group mechanisms, attention and resilience. The rhythm battle also espouses the ten characteristics of CDI we listed in Table 1. While the overall structure of the game is provided from outside, the game is open-ended (2), and while the groups are given a tempo for their rhythmic movement, the solution space is very large, as groups can freely improvise rhythms using any metrical arrangement and any body parts they wish. The structure of the game is designed with characteristic (8) in mind especially. Shifts from resisting entrainment to “falling” to the other team’s tempo are bifurcation-like [63], and the likelihood of changes in the tempo dynamics within and between teams is increased by having the teams move in space closer to each other.

A growing body of literature suggests that moving in a synchronous manner enhances social bonding (e.g., [64,65,66]). We were interested to find out to what extent the rhythm battle task had an impact (or at least correlated with) the strength of felt relatedness among the group of participants. We wanted to investigate (i) To what extent participants felt more related to members of their own rhythm group compared to the members of the other group; and (ii) To what extent the relatedness or felt proximity between participants was related to the similarity of their tempos.

To measure the felt relatedness, we developed a novel task based on the Inclusion of Other in Self test (IOS, [61]). In the original IOS test (developed for romantic partners) the participant chooses which of the seven Venn diagram-like pictures (where one circle represents the participant and the other their partner) best represents their relationship. The diagrams range from no overlap to near complete overlap. In our task, participants used tablets, and were asked to draw themselves as a circle, on a background where the circle representing another participant is already drawn. Thus participants can represent interpersonal distances that are outside the scale of the original test, which makes the test more adequate to describe relationships with people they have just met in the dance improvisation experiment, for example.

#### 6.2.1. Methods

*Participants:* In a pilot study, we analysed data from three groups of 10 participants. Each played the game once, as a part of a dance improvisation class, and all had some experience in dance and/or music. The groups had 10–14 participants, but only ten in each group wore the accelerometers that provided our data. Of all the participants, 18% were men, 55% identified as amateur dancers, 25% as professional dancers (representing various forms of dance), and 21% indicated no previous experience in dance. The amateurs had an average of 10.5 (SD 8.9, range 2–30) years of dance experience, while the professionals had an average of 17.3 (SD 6.5, range 11–29) years of experience.

*Equipment and measures:* Participants wore BioHarness accelerometer pods on chest straps. We calculated the periodicity, or tempo of the movement, based on peak-to-peak intervals, in each of the stages of the game (see protocol below). The tempo differences for each pairwise combination of participants (both within and between teams) were then calculated, and assembled into tempo similarity matrices (see Figure 2C).

After the experimental session that contained also other in sport, activities and tasks, participants were asked to rate their closeness to each of the other participant, using a tablet-based version of the Inclusion of Other in the Self-measure (IOS, [61]). Rank-order scores of these ratings were averaged for each participant pair (e.g., the closeness rating score for participants 1 and 2 was the mean of p1’s rating of p2, and p2’s rating of p1). The matrices of these ratings were compared to the tempo similarity matrices.

The correlation analyses using tempo similarity matrices, team membership matrices, and closeness rating matrices were conducted using Mantel tests, where the significances of the correlations were estimated using permutations with 10,000 iterations [67,68].

In the network analysis, we convert the tempo similarity matrix into a graph so that each participant is a node, and the weights of the edges (connections between the nodes) represent the tempo similarities. The closer two participants are in tempo, the stronger their connection. The network visualisation was conducted in Gephi 0.9.2, using a Force Atlas algorithm that pushes weakly connected nodes away from each other and pulls strongly connected nodes closer.

*Protocol:* In each group, the participants were divided into two teams. Each team had a facilitator with a metronome. The two metronomes were set to different tempi (85 and 100 BPM). The game has four stages (Figure 2A).

In the first stage, the two teams were on different halves of the room that was divided using a movable wall. Teams were given a task to create a group rhythm pattern that they could then move to. Participants could use any part of their body for generating the rhythms: clapping hands, stomping feet, snapping fingers, etc. They could decide how complex they wanted to make the rhythm, but they had to perform it in the given tempo.

After the teams figured out their patterns, and started continuously performing them, the room divider was removed, and thus the teams were now coupled to each other. At first, the metronomes were still on, so that the teams could get used to the distraction of the other team and have a clear referent of their original tempo at hand.

In the final stage, the metronomes were turned off, and the teams were asked to start moving in space, as groups, still performing their rhythms. They were given an objective to move to the other end of the room, requiring them to pass the other team at close proximity. The game took about 10 min in total to complete.

At the end of the dance intervention (which the rhythm battle was a part of), each participant completed the tablet version of IOS. The participants were all sitting in a circle with their number tags clearly visible, and each participant rated their closeness to every other participant, as prompted by the tablet.

#### 6.2.2. Results and Discussion

As the objective of the game was to maintain the team’s original tempo, we could look at what tempi the participants moved at the end of the game, and see which team had “won”. In all of our three games there was a “winner”, in other words, in none of these cases did both teams manage to maintain their tempo until the end. We observed two different ways of “losing” the game. Either the winning team managed to maintain their original tempo, and get the losing team to entrain with them and drift in tempo until everyone in the game was moving at the same tempo. The other way of losing was to lose the team’s internal coherence completely, often with a subset of the losing team entraining with the winning team, and other team members ending up with either very variable movements tempi, or tempi outside of either team’s original tempi. An example of this second type of loss was observed in the second group (Figure 2B).

We used multiple ways to look into how the games progressed towards these outcomes. First, we looked at how the tempo differences that we observed matched with those expected based on the team membership, in each of the stages of the game. At the start of the game, the tempo similarity matrix correlated with a group membership matrix ($Spearman^{\prime }s\;rho=0.43,p=0.03$) (Figure 2D). The tempo similarity matrix for the start of the game (Figure 2C) shows there were two distinct teams. The tempo differences within both teams were small (indicated by bright colours), while differences between teams were larger. There is one outlier, participant 3. Also the mean tempi (Figure 2B) show that the performed tempi were same as designated by the metronomes.

As the game progressed, the correlation between the tempo similarity and team membership matrices got lower, and by the end dropped to zero, indicating there no longer were two separate teams. The average tempo graph (Figure 2B) shows that only the faster team maintained constant tempo, and a constant tempo variability, as indicated by the error bars. The slow team lost their within-group coherence, as indicated by their large error bars. The tempo similarity matrix confirms that some participants failed to resist entraining with the fast team, while others became outliers, not close to anyone’s tempi.

The network visualisations (Figure 3) provide another way of examining the tempo relationships between and within the teams. These relationships can be visualised as a dynamic network, but here we present only two static snapshots, one from the first stage and another from the final stage. These networks are based on tempo similarity matrices (Figure 2C), and their main contribution here is that they provide a visualisation that allows us to see what happens in the pairwise relationships. In the beginning (Figure 3, top panel), apart from one outlier, the graph looks exactly as we’d expect for this first phase: two groups, each highly coherent within, and with a clear difference to the other group.

Figure 3 (bottom panel) shows the network at the end of the game, and the visualisation confirms why the slow team lost. Its participants had gone their separate ways, one ending up with the outlier from the fast team, another becoming part of the faster team, isolating the other two. The fast team was stronger and won, after the slow team lost their internal coherence and split up.

We looked at whether the closeness measure correlates with the group membership. Pooling together ratings from three groups of participants (three games), we compared whether on average, members in the same team were rated closer than members in the opposing team. Indeed, a Kruskall-Wallis test indicates that own team members were rated as being closer than participants in the competing team ($\chi^{2}=7.5,p=0.008$).

We also wanted to see whether the closeness rankings obtained from the IOS were correlated with observed tempo differences, i.e., the actual amount of synchronisation during the game. Figure 2D shows that they were not in the beginning (indeed, the tempo differences at this stage were instructed by the experimenters), but towards the end of the game, where metronomes were off and the tempo differences could freely change, there was a trend for a correlation in the second group. This result invites speculation about the direction of influence behind this correlation: did participants who synchronised with each other in the rhythm battle rate each other as closer due to this encounter, or did they originally synchronise because they had a closer relationship already during the game? Or, was there another factor that influenced both these variables?

At the end of the session, the participants filled a questionnaire regarding their experiences. One of the questions concerned their enjoyment of the rhythm battle game (on a 1–5 scale). In order to asses the relation between the tempo differences between group members and this enjoyment-related facet of their aesthetic experience during the game, we correlated (Pearson’s r) their responses with three variables: the (absolute) distance of their individual tempo from the group’s mean at the first phase, the (absolute) distance of their individual tempo from the group’s mean at the last and the absolute tempo change between the two phases. The negative correlation between enjoyment and last phase distance from group mean was significant (*r* = −0.73, *p*-value = 0.026), so the closer the participant’s tempo was to their group mean, the more the person enjoyed the game. The same held already for the first phase but the correlation was not quite statistically significant (*r* = −0.63, *p*-value = 0.066). Finally, a similar non-significant negative correlation was found between the change in tempo from first to the last stage, and enjoyment (*r* = −0.62, *p*-value = 0.076). These findings suggest that the more stable one’s tempo was throughout the game, the more they enjoyed the game. It is important to note that the three measures computed from the movement data were correlated so it is not possible to propose a firm interpretation. What we can conclude is that third person measurable performance on the task (maintaining a group tempo) was correlated with a first person experience of pleasure.

## 7. General Discussion

### 7.1. Mirror Game

Based on our results and experiences during the pilots we consider the four-person mirror game as a rich and valid protocol for the study of non-verbal communication and group dynamics. The game is accessible, intuitive and engaging for both improvisation experts and novices. While the existence of multiple strategies or conceptualizations of mirroring in this version makes the task closer to the experience of free improvisation, the movement patterns remain relatively restricted and tractable, allowing the application of existing analyses and measurements. Our results demonstrate that the four-person mirror game can be used to probe differences in intersubjective tuning or coordination following even very short interventions, differences that might not be able to be captured by other existing measures.

This is, to our knowledge, the first published attempt to extend the dyadic mirror game paradigm to larger ensembles. The most notable result of the Noy et al. (2011) paper, which has been replicated in different follow up studies [13,69,70] has been the particular kinetic signature of the no-roles condition, named often *co-confident motion*. When expert dance improvisers played the game after an intervention specifically priming coordinated motion (pilot 1), we observed the emergence of highly correlated, smooth motion that we consider to be the signatures of the same co-confident motion. However, different kinetic patterns emerged when non-experts played the game after an intervention that did not prime coordinated motion but rather turn-taking (pilot 2). Based on combining the kinematic data and participant interviews, we identified at least two different strategies or patterns of coordination: rotating leadership among all four participants with dynamics reminiscent of conversational interactions, and splitting of the group into two dyads.

We take the diverging results as both a promise and as a challenge. The promise is in the apparent sensitivity of the game to differences across populations and interventions. The challenge is to carefully investigate (at least) these two parameters so to be able to confidently relate them to different emergent strategies in the game. In ongoing work we are looking at musicians with different levels of expertise and evaluating the impact of two highly similar joint musical improvisation tasks, one favoring simultaneity and one favoring turn taking. We also suspect that the exact wording of the game instructions has a significant impact on the choice of strategy. The results of the second pilot highlight the importance of integrating qualitative analyses (from both player and spectator perspective) with the quantitative movement measures. There was an indication that participants’ personality traits, in our case trait cognitive empathy, might influence their behaviours during the games. We are continuing to develop means to collect detailed first person reports both in real time and post-game.

### 7.2. Rhythm Battle

The group rhythm battle, based on our pilot results, is a promising ecological task which is intuitive, engaging and appropriate for novices. Importantly, the task was embedded in a dance improvisation workshop and was experienced by the participants as such. Six groups of non-experts have been able to co-construct a joint rhythmic pattern which they followed and maintained (to varying degrees) with a common tempo after the cessation of the metronome. When the two teams were allowed to interact, groups and individuals exhibited different degrees of entrainment and entrainment resistance. This variability makes the task a promising tool to evaluate both individual sensorimotor synchronization capacities as well as group attunement and resilience. Group attunement can be estimated from the overall synchronization of all its members (particularly in the first part of the task). We identified two measures of group rhythmic resilience: the extent to which the group maintains its original tempo, and the extent of the increase in variability among group members’ tempo. In the three battles conducted in the pilot a clear ‘winning’ group always emerged, following one or both of these criteria.

Group membership in the rhythm battle was significantly correlated with the sense of affective proximity that participants felt between one another. We also observed a trend of a positive correlation between tempo proximity and affective proximity at the final segments of the rhythm battle. While we cannot infer from these results that the rhythmic task was necessary for the sense of closeness to emerge (as they could just be effects of being grouped together, cf. [71]), they serve as a double validation. These results suggest that the task is at least sufficient to induce of a sense of proximity, and it validates our tablet-based version of the IOS as a test of felt proximity.

In addition, the negative correlation found between one’s tempo distance from the group mean and the pleasure experienced during the game provides a preliminary evidence for a relation between togetherness (or at least a third person measurable proxy of it) and the aesthetic experience of the participants. While enjoyment is arguably a component of aesthetic experience, the latter is not limited to it. In future research we intend to include additional dimensions of this complex intersubjective experience.

Previously, group synchronisation has been studied relatively rarely, but recently studies have looked at group-level behavior in electronic dance music (EDM) [72,73], as well as people dancing in silent disco [74,75]. In contrast to ours, the silent disco studies look at outcomes (pro-sociality, memory) at individual levels. The EDM studies characterise collective movement parameters in the specific musical context, and [73] utilises both first person and third person views, emphasising how embodied and interactive the EDM experience is. The main difference between these studies and ours is that in the rhythm battle the participants supply the rhythm themselves, whereas in the EDM and silent disco studies the music and the rhythm comes from outside, and the participants dance to these external rhythms. However, especially the EDM research directions are relevant and interesting for our future work.

The empirical work reported here is the first step in our goal to address the theme of this special issue, embodied aesthetics and the specific thesis of this article, namely togetherness as the embodied aesthetics of joint improvisation. We have established (or at least provided a proof of concept for) a set of devices that generate, in a semi-ecological setting, coordinated interpersonal behavior that can be quantified and modeled. Though both devices described in this paper were constructed based on the hypothesis that togetherness is translated kinetically as simultaneity, it seems that this vision of togetherness might need to be revised. It is possible that other behavioral coordination strategies (such as spontaneous turn taking) can also be expressions or causes of togetherness. We will return to this point in the conclusion.

The variability both at the levels of individual and of group behavior, and the apparent sensitivity of the observed behavior to a multiplicity of factors, is an expected feature of an ecological setup. However, much work is still required in order for these tools to become fully functional.

We have demonstrated that the behavior produced by these devices is sensitive to training and experience and so they can be used as probes. In addition, we developed and tested a measure of group connectedness based on the IOS test. This tool provides a post-hoc measure of affective proximity between group members and we have shown that it can track the impact of at least one of our devices. However, the relationship between post-hoc affective proximity and felt togetherness during the dance practice itself is an open question. In order to go further towards the goals of our project we plan to implement methods to collect first- and second-person reports of togetherness either in real time (using custom built tablet-based technology) or using post-hoc video annotations. We also plan to use physiological data (e.g., ECG, EDA) and subjective reports to find out the psychophysiological and affective correlates of felt togetherness.

As these protocols have been born from the interactions of dancers and scientists, they combine ingredients from both sides. While they have features that “tick the boxes” of the design features of ICI protocols (Table 1), they can be further adapted to either artistic or scientific needs in mind. For example, both the four-person mirror game and rhythm battle can exist in different versions in the future, either as starting points or inspiration for open-ended collective improvisation scores, or as even more carefully controlled experimental tasks. In the versions presented in this paper, the balance has been somewhat on the scientific side, as in both cases we have prioritised collecting relatively well-controlled data over allowing as much creative freedom of improvisation as we typically would. This is because at this stage, we need data from e.g., motion capture system (that poses many constraints to the movements of the participants and the length of recordings we can make, compared to our usual setups) for the development of measures and models that characterise these behaviours and group-level interactions. After having figured out these measures, we can implement them in a more naturalistic context, using simpler wearable sensors and less constraining protocols.

## 8. Conclusions

In this paper, we addressed the issue of togetherness in its kinaesthetic dimension. Kinaesthetic togetherness, the joint feeling of moving with and being moved by others and the whole group we form, refers to both an observable phenomenon from an external point of view (the point of view of spectators or scientists) and a qualitatively lived experience from a first-person point of view. The investigation of this phenomenon therefore requires to overcome the gap between these perspectives [76]. Our project is an attempt to build a bridge that fills that gap.

We hold that a strong foundation for such a bridge can be found in collaborative artistic practices. For example, in collective dance improvisation the interplay between the feeling associated with moving collectively and the observation (from the inside or the outside) of such a movement coordination is already there, and advanced practitioners are experts in its handling. Such contexts are therefore appropriate for the intellectual and empirical investigation of kinaesthetic togetherness, and a dialogue with the expertise of practitioners can guide this endeavor. The second, theoretical, foundation of our approach is the enactive and dynamical view of cognition which provides a conceptual framework and formal tools to connect first- and third-person perspectives, as well as an operational definition of interpersonal dynamics. Our working hypothesis is that togetherness manifests itself at the conjunction of first- and third-person perspectives, and that this is possible thanks to the dynamics of the interaction between individuals involved in the co-regulation of their collective effort (first person plural). Furthermore, we propose that this emergent phenomenon as well as its felt experience are aesthetic in nature. A similar view has recently been discussed in the area of collective or shared emotions [19], and exploring the affective processes and group-level emotional mechanisms will be important in our project going forward.

One central methodological principle that follows from the above-mentioned considerations is that the research process itself must be embodied or lived. Rather than bringing dancers to the lab and observe and interrogate them, we bring the scientists onto the dance floor thus colliding theoretical assumptions with lived experience.

In our experimental scores (two of which presented in this paper) we examine how interactions bring about movement coordination between groups of people moving in space according to various rules. We also examined how interventions such as guided interactive improvisations could enhance such coordination and how they impact first-person experiences that are related to one’s sense of connectedness to others. While the studies discussed here concern mainly the observation of behavioral correlates of togetherness as they can be measured from an external point of view (third-person), the tasks themselves emerged through first-person and collective experience of improvisation in the research group.

In the mirror game, we observed differences both in the degree of coordination itself, and in the strategies deployed to complete the task (mutually shared motion versus turn-taking). Not only did the task enlighten our comprehension of the possible ways through which people act *together*, but it also showed how first-person reports can guide the analysis and understanding of interaction dynamics (as in [77]). More experienced improvisers were able to perform co-confident movements, and reach a state of shared agency and togetherness, where movement emerges without anyone overtly initiating it. In the one-dimensional mirror game, as in [55], it was also found that only experienced improvisers could enter the co-confident motion state. Interestingly, [13] did not find this in their two-dimensional mirror game, but instead observed co-confident motion in dyads of novices. The novice groups in this study shared leadership in a way that does not indicate shared agency, and based on the post-experiment interviews, did not seem to result in a particularly strong feelings of togetherness. There is no reason to assume a priori that synchronicity automatically leads to experiences of togetherness, or that it would be more likely to do that than an interaction based on other coordination modes, such as turn-taking. However, in this case, the strategy to share leadership by taking turns to (trying to) lead the movements was associated with confusion, as having to figure out what movements to make was compounded by having to figure who has the authority to lead them and suggest new patterns. Thus participants probably lacked the sense of having achieved their joint goal, which would influence their experienced we-agency [19].

The situational factors that contributed to the novice mirror games not leading to togetherness should be investigated in future studies. It would be interesting, and relevant from pedagogical point of view (see e.g., [78]), to figure out what type of induction would allow novices to tune in to each other in these larger groups to such an extent that they’d be able to produce emergent movements under a shared agency.

The rhythm battle task has proven to be a good task for observing group entrainment, as it superposes within-group entrainment task with a task to resist entrainment to the other team. As it is also an improvisation task, it provides a much more ecologically valid context for studying entrainment than e.g., tapping tasks do. Participants need to genuinely balance between producing aesthetically pleasing, varied and complex rhythmical patterns, and the “pragmatic” need to focus on a simple, easily entrainable beat. This balance needs to be achieved through a collective regulation of the group’s dynamics. We also observed that the participation to this task entailed a feeling of bonding with members of the in-group, although our study design cannot distinguish whether this increased feeling of closeness with in-group stems from being assigned in the group and having the shared experience, or whether the closer synchronisation that in-group members experienced, had an effect. Furthermore, pleasure during the game (an aesthetic feeling), was correlated with the extent that the person’s tempo was similar to that of their group (which is a potential dimension of togetherness).

Admittedly, we have not yet directly targeted the first-person dimension of togetherness by our empirical designs, and one of our future challenges is to develop better methods for capturing the subjective experiences of improvisers, and combining those with the third-person data. The emergence of various forms of strategies of co-regulation also poses the question, whether these different strategies could all lead to such a feeling? In other words, under what conditions can non-simultaneous, turn-taking-like forms of co-regulation lead to or be considered as togetherness? We have yet to test whether interaction dynamics and their active co-regulation are both sufficient and necessary to bring forth the phenomenon of togetherness from the joint perspectives of first- and third-person points of view.

Noy et al. [79] explored togetherness in a mirror game, comparing first person reports with third person kinematics, and found that they correlated mildly, as participants’ subjective indications of togetherness exceeded those that could be identified from the kinematics.

In dyadic situations, synchronicity leads to prosociality and increased empathy etc. [64,66]. In the context of a dyadic mirror game, high coordination correlated with coordination of heart rates between the participants [79]. Recently, this question about links between synchronisation and pro-sociality, was posed about groups [80]. Zimmermann et al. found that while group synchrony did not predict pro-social behaviour, the distributed coordination in the group predicted how much participants liked each other, their feelings towards the group, and the conformity of their opinions.

Another dimension that we plan to address in future studies is the affective dimension of interactions, especially the emergence of collective, shared emotions [19]. Joint affective processes are intrinsically linked to the embodied interactions and as Salmela and Nagatsu point out, central to the shared agency. Our combination of first and third person perspectives is also important in uncovering the affective dynamics in groups.

In the context of this discussion, it is interesting to note that [12] found that spectators’ annotation of the extent of togetherness (moving together) during a group improvisation performance did not correlate with accelerometer-based measure of movement synchrony. One possible interpretation of this result (though see the cited paper for other directions) could be that the subjective sense of ‘moving together’ is not a (simple) function of observed kinetic synchrony but might include other factors such as perceived shared intentionality [81]. We also have to provide further empirical support for the idea that this conjunctive phenomenon is aesthetic in nature. Vicary et al. found that, at least in certain performances, movement synchrony was predictive of the spectator’s enjoyment (their index of aesthetic experience) despite not correlating with perceived togetherness. Perceived togetherness was correlated positively, in some performances, with enjoyment. This result suggests that perceived togetherness and movement synchronicity can both contribute (independently) to the aesthetic experience of a spectator. Our results from the rhythm battle go in the same direction (but from the dancer’s perspective this time): tempo similarity was correlated with enjoyment of the game. To explore this further, our primary challenge is to refine the methods through which we can collect first-person experiences of togetherness and aesthetic experience from the dancers themselves (in real time or through post-improvisation video confrontation). Our protocols do not divide participants into performers and spectators, which reflects our understanding that the aesthetic experiences are inherently collective in nature.

In this paper, we have presented ideas and observations to argue that togetherness is a phenomenon that can be approached through the interactive conjunction of practices and perspectives. Our proof-of-concept studies encourage us to pursue the hypothesis according to which playing with togetherness kinaesthetically can be a source of collective, aesthetic feelings. Our approach and these findings can be useful for the study of social psychology, and especially the studies on the links between synchronous behaviours and their positive affective and social effects. We argue that a direct link might be too simplistic, and exploring the mechanisms behind these effects would benefit from re-framing the question to be about togetherness, and acknowledging the role of managing synchrony *together*, a meta-level of coordination usually absent in experimental studies of behavioural synchrony. One future direction for project ICI is to explore scores that focus on non-synchronous interactions and other modes of coordination, where synchronisation of the participants is not instructed, but rather, might spontaneously emerge from their interactions. These more open-ended protocols will benefit from the measures and methods developed in the more experiment-like protocols presented in this paper.

Our study, and the enactive, kinaesthetic and collective perspective on the aesthetic experience of group dance improvisation developed in the introduction, can inform the fields of art research and aesthetics, and in particular the sub-field of empirical aesthetics. Improvisers and improvisation teachers in all domains could benefit from the bridge we make here between practice and theory, regarding the mechanisms at play during improvised interactions. Dancers and choreographers could use these mechanisms as inspiration, or explore these mechanisms and their effects through new scores and works. Most of all, we hope that our example of collaboration between artists and scientists would encourage others to take on similar collaborative projects.

## Figures and Tables

**Figure 1 behavsci-08-00023-f001:**
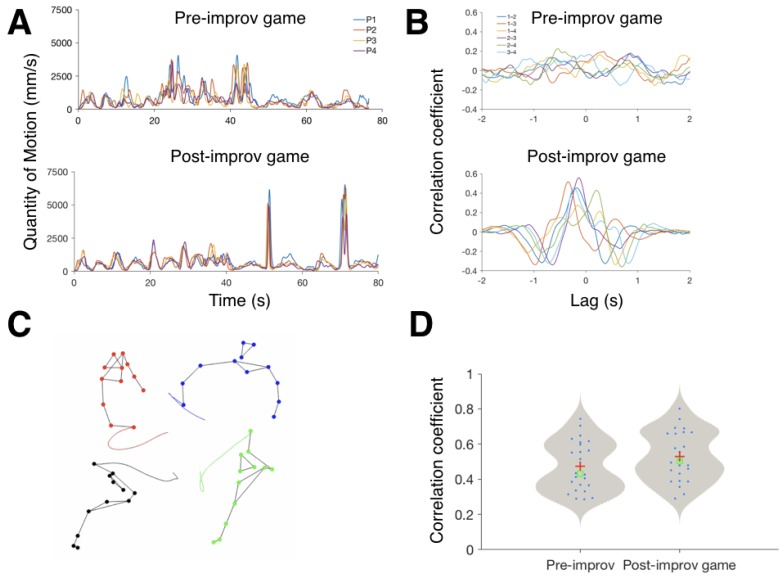
(**A**) Quantity of Motion in pre- (top) and post-improvisation mirror games in pilot 1; (**B**) Cross-correlation functions in pilot 1, indicating that group-level coordination was absent in pre-, but present in post-improvisation game; (**C**) Participants were standing in a circle, one arm and index finger extended. Frame from pilot 1, the movement trace shows finger trajectories for 1 s; (**D**) Distribution of maximum pairwise correlation coefficients in pilot 2. Red crosses are means, green dots medians, and blue dots represent the data points (pairwise CC’s).

**Figure 2 behavsci-08-00023-f002:**
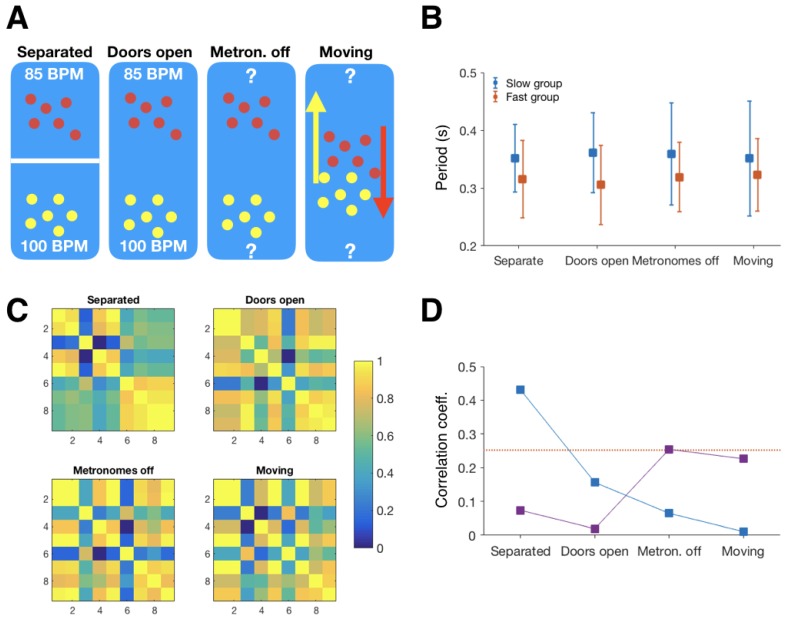
(**A**) Structure of the rhythm battle protocol. (**B**) Average tempi in each of the segments. Data from the second group. Error bars represent averages of individual participant’s tempo variability. (**C**) Tempo similarity matrix from the second group. Yellow represents high similarity and dark blue dissimilarity. Values are normalised. (**D**) Correlations between tempo similarity matrices and group membership (blue), and tablet-IOS scores and tempo similarity (purple). Red dotted line indicates the p=0.05 threshold. Data from second group.

**Figure 3 behavsci-08-00023-f003:**
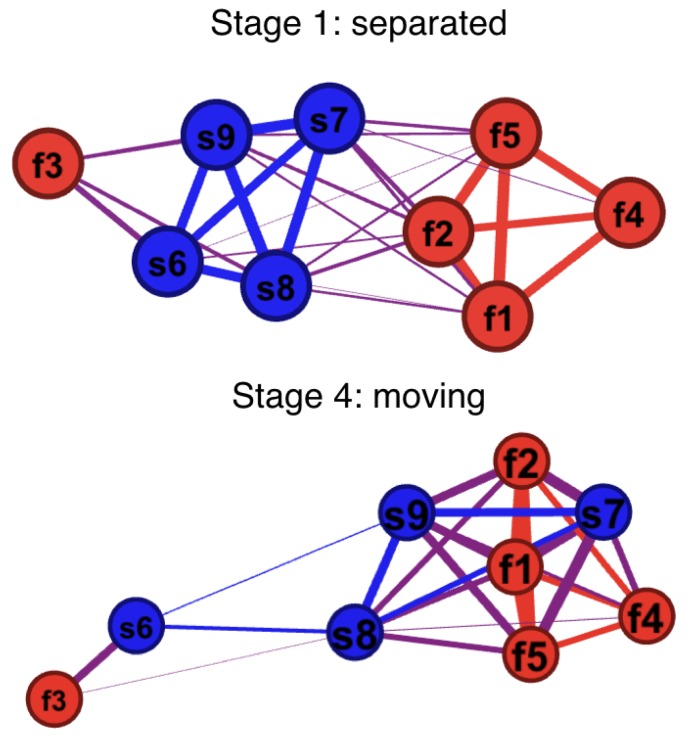
Network visualisations of the tempo similarity matrices in stage 1 (**top**) and stage 4 (**bottom**), in the game for second group. Node colors indicate group membership, thickness and also length of the edges reflect the tempo similarity with thicker and shorter edges reflecting closer tempo similarities.

**Table 1 behavsci-08-00023-t001:** The ten design features of ICI protocols.

1	Design through experience
2	Open-endedness
3	Poly-solution
4	Interactive generativity
5	Always more than two
6	Measurability
7	Experience-independence
8	Identifying and eliciting phase shifts
9	Primacy of kinaesthetic experience
10	Fun/Pleasure

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
