# Peer review of "Coordinated Interpersonal Behaviour in Collective Dance Improvisation: The Aesthetics of Kinaesthetic Togetherness"

_behavsci, 2018, doi:10.3390/bs8020023_

Round 1

Reviewer 1 Report

Dear authors,

thank you for this interesting piece of research

I would suggest to enrich your paper dedicated to improvisation considering the following papers:

Biasutti M. (2017). Teaching improvisation through processes. Applications in music education and implications for general education. Frontiers in Psychology. 8:911.

http://dx.doi.org/10.3389/fpsyg.2017.00911

Biasutti, M. (2016). Variables influencing improvisation: educational implications. In M. Santi & E. Zorzi (a cura di) Education as jazz. interdisciplinary sketches on a new metaphor (pp. 159-176). Cambridge: Cambridge Scholar Publishing (ISBN-13: 978-1-4438-9070-0; ISBN-10: 1-4438-9070-7).

Biasutti, M. (2013). Improvisation in dance education: teachers views, Research in Dance Education, 14 (2), 120-140. DOI.org/10.1080/14647893.2012.761193

I addition, for guiding the readers, it would be nice to have the research questions of your study.

In the discussion please conncet better your research results with the literature

Please add some limitation for your study because you were testing only few participants

Author Response

Our response to all reviewers is attached in the PDF. In summary, we rewrote much of the introduction and added a section to tie the background and empirical work together. We also added references to intro and to conclusions. 

Reviewer 2 Report

The paper convincingly and successfully brings together discourses from phenomenology and affective approaches to movement.  Excellent use is made of sources such as Simondon, Manning and Stern. It also makes very good use of existing research on synchrony and entrainment.

It is very clearly organised and the descriptions are full and clear. It is very well written.

The arguments about individuation, blending of agency and embodied togetherness are very well arrticulated and are convincing.

The paper’s title refers to ‘collective dance improvisation’ and the abstract contains references to dance forms. What is meant here by ‘traditional’ and ‘social’ dance? The abstract also refers to contact improvisation. Here there are no set steps and the dance is created at the same time as it is performed, through the physical contact of dancers responding to each other’s movements and weight. Standard texts on this include Novak, Cynthia J. 1990, Sharing the Dance: Contact Improvisation and American Culture. Madison and London: University of Wisconsin Press; Taken by Surprise: A Dance Improvisation Reader,  Wesleyan University Press 2003, edited by Ann Cooper Albright and David Gere.

Since the paper does not develop these aspects perhaps they should not be given prominence at the start. Perhaps this could be developed further in the future.

Sometimes it isn’t clear how the word ‘aesthetic’ is being used – what is the difference between an experience of togetherness and an aesthetic experience of togetherness?

The tenet that research processes must be experienced and that scientists can be brought onto the dance floor rather than dancers being confined to the lab seems an excellent way forward.

The second experiment in particular, the ‘rhythm battle’, was very well designed to reconcile ecological demands with the constraints of measurability. This is also methodologically significant in terms of correlating first person and observable experience.

Author Response

(The authors gave the same response as above.)

Reviewer 3 Report

This work is sound and compelling.  The article describes and analyzes the studies conducted with thoroughness and readability.  

That said, I'd feel that the two halves of the paper feel somewhat disconnected.  The first half, which provides context, defines terms, reviews literature, and suggests hypotheses seems, to me, to promise a somewhat different project than what is described in the second half of the paper.  The second half of the paper, as I state above, describes and analyzes the protocols performed quite clearly, but they seem like protocols for a different set of questions.    

The first half suggests a notion of togetherness without explaining, until towards the end of the paper, that togetherness is measured in the studies via synchronized movement.  This is somewhat misleading after the discussions of vitality affect, etc, which don't specifically define togetherness as synchronicity.  

Also, the first half suggests that the authors have conducted numerous verbal interviews with the participants, but the second half does not provide any of the material gleaned from these interviews.  

Similarly, the abstract mentions a phenomenological perspective whereas I don't see how the protocols get at any kind of phenomenological perspective.  Also, kinesthesia is also mentioned, but  the protocols don't really seem to approach a study of kinesthetic experience. 

I'm not totally clear on the authors' use of the idea of "aesthetics."  Is it necessary to use this term?  It doesn't seem appropriate to describe the improvisation studies that they conduct.  

A little more clarification on how the authors are using an idea of participant response versus observer response would be helpful.  It seems to me that the studies described are all based on outside observation.

I think that the improvisationstudies performed are interesting and rich.  Perhaps the authors' could reframe the first half the of the article to better contextualize what the research truly asks and answers?  

Author Response

(The authors gave the same response as above.)

Reviewer 4 Report

This is an interesting article/paper coming out of a study that proposes more broadly that improvisational dance can be measured for perceptions of kinaesthetic ‘togetherness’. The authors propose two new methodologies to measure kinaesthetic togetherness of improvisational dance, which they also refer to as ‘games’. 

The paper is most promising in the sections that discuss dynamic systems theory and its relationship to togetherness in dance improvisation. This is an area of research where much work could be done in the field and pages 7-8 are particularly good in the ideas and the potential relationship to dance improvisation practices. The relationship to aesthetics here would be particularly interesting as well, for instance how dancers might perceive they are dancing together in a way that matches their expected aesthetic desires or not. It seems that Barbara Montero’s 2006 paper on 'Proprioception as an Aesthetic Sense' would be useful, but it is not cited. 

However, given the sections in the setting up of the paper, the studies that are shared do not meet these ideas as was expected and it is still unclear if/how the paper is set up in the first half is as useful to the methods and data that the authors share in the second half of the paper and whether the authors can clarify the overall aim of the paper, and then in turn the discrete parts to meet that aim.

There is a part in the paper that the authors state they want to establish their methodologies ('global methodology') in this paper. This seems to be promising if sticking with the second half of the paper, clarifying more of the details of what was done and discussion of how methods might be validated and why the choice of what is measured is important, e.g. why is measuring synchrony by motion capture data useful to the authors’ definition of togetherness; and do the authors care if there is or is not a relationship to what they perceive of motion capture data of 'togetherness' and the dancers’ experiences, since the dancers might have not felt togetherness in this particular exercise (we are not told if they did or did not)?

There are many questions about what kind of dance improvisation the authors refer to throughout. Contemporary dance improvisation? Or is it assumed that it does not matter what style of dance improvisation and that they are homogenous? There is much research in dance studies to counter the latter. Related to this it seems to be assumed that dancers can be grouped homogeneously as well. There is no information about the difference in and types of training of participants that are ‘expert’ dancers, or ages and genders of participants and that this information does not inform the measurement of kinaesthetic togetherness, which raises some questions.

Throughout there are also questions about whether the paper is trying to cover too much and a key part of this is whether the authors want to discuss games or dance experience, since the two studies included are rather different in relation to each other. The attempt, within a considered word count it is understood, to address the relationship is noted, however the two studies are not both dance; it might serve the paper better to choose only one of the studies for the paper so that the rest of the paper can have better focus. 

And on the subject of games, and referring back to the issue of difference regarding participant traits, Pierre Bourdieu’s use of the game analogy in the theory of habitus came to mind as well. How does that relate or not to what the authors are proposing about the relationship of togetherness and games? Again I refer back to what seems most promising of the article, the issue of dynamics systems theories and how participants might need to follow similar rule structures to then allow something else to happen together, and that this might be brought out more somehow in revisions.

In the beginning of the paper, the idea of acting, deciding and solving and its relationship to agency are interesting and make sense in relationship to kinaesthetic togetherness for the mover. How does this relate to the studies that are conducted and shared however?

There are several statements made that are opinions rather than based on sound and rigorous research to form an argument. Examples (beginning of sentence provided for reference): p 1 ‘ there seems to be an intrinsic pleasure involved…’ and ‘Collective dance improvisations seem to be based…' p 2 ‘because contemporary artists are more and more involved in shaping… p 4 ‘ the experience of moving together thus brings the subjects to a non-dualistic…’ (how can we know this statement made here? what is it based on?) p 5 ‘the dominant view in cognitive science describes social  interaction as a succession…’ (This section presents a limited review of cognitive science today and it might be better to start at the dynamic systems sections for the beginning of the paper instead, around p 6) p 6 ‘in a collective dance improvisation , individual activities…’ (it is difficult to universalise dance experience in this way)

p 4 ‘our understanding of collective dance improvisation…’ - here, as mentioned above, what kind of dance, by whom, how can this statement be presumed for all types of dance, dancers, and contexts?

‘On a more phenomenological level’ implies some simplicity about phenomenology, which the authors probably do not mean to do. Greater depth of the phenomenological perspective used for the study would be useful.

There are several hints at looking at qualitative data of kinaesthetic togetherness such as on page 11, page 16, and 17 and it is not clear why this is not done for this paper - the sharing of qualitative data from the studies - as it seems it would fulfil some of what was expected after reading the set up to the paper. It is difficult to follow how the studies without qualitative data, i.e. whether the participants experienced kinaesthetic togetherness, are useful and why it is crucial that these studies get published separate to that qualitative data. Again, it seems the authors are possibly trying to establish a ‘global methodology’ however the aim of the article to do that, the discourse used to set it up (first half of the paper) and precisely which of the studies are shared and why, can still be worked out to make this much more clear.

Author Response

(The authors gave the same response as above.)

Round 2

Reviewer 1 Report

thank you for the kind revision. 

I am sorry but I am not able to verify if all the change were made because the track version was not used. I gave a quick look and I I noted that not all the literature that I suggested (which contained specific papers on dance improvisation) was included. I would be grateful if you could send me a track version of the paper.

Author Response

Unfortunately, as we’ve worked on LaTeX we can’t easily provide a track changes -version of the two versions of the manuscript. In addition, it would probably be more confusing than clarifying, as especially the introduction and conclusions were very heavily reworked, with changes in or complete rewrites of every paragraph, including re-ordering the paragraphs. Section 5 (Investigating togetherness: the ICI project) is mostly new text, while in the empirical sections, we made only smaller changes, to tie these sections better with the background, and the new section 5.

Specifically to suggestions of reviewer 1, we’ve added a reference to Biasutti et al. 2017, on line 886 (reference 72).

Reviewer 3 Report

introduction much clarified and made more directly relevant to the protocols described in the second half.  everything tied much more closely together.  terms more clearly defined and deployed more consistently throughout.  

Author Response

Thank you again for your comments and suggestions, and helping us make the paper better.

Reviewer 4 Report

This is a much improved paper and I commend the authors for the response and turnaround. There is still a lot, sometimes it seems too much, covered in the paper, which make it difficult to encapsulate an argument more succinctly. The emphasis of the work as a series of pilots and set of devices, in a semi-ecological setting, to quantify and model coordinated interpersonal behaviour is more representative of the work and an improved framing. The introduction is also much improved and sets up the paper better than the previous draft. The writing and clarity of the methodology and findings in this version allow one to understand what was done better and follow the innovations and interdisciplinary work that the authors are doing and trying to achieve. The work is in parts informative to contemporary dance improvisation pedagogy and practice and ways of thinking about kinaesthetic togetherness.

I have a few comments from reading this version, that the authors can respond to as they wish or not-

The second pilot - the rhythm game - it is stated on p 18 (around 713) that the participants enjoyment was the main measure of aesthetic experience. However, my impression for the mirror game and the previous description of aesthetic experience related to kinaesthetic togetherness, that  aesthetic experience was limited only to enjoyment. So there seems a disconnect there between the thinking behind the studies and then what you are measuring and commenting on for this second pilot. This might be an inherent challenge of working with rich material and in the dance environment, rather than in a more strict scientific paradigm and experimental situation, in that the complexities take a great deal of attention, explanation, and consideration when then going back to the resulting 'data'.

Framing your 'findings' as 'strategies or patterns of coordination' (p 18, line 747) seemed the strongest way of referring to what you are doing in the paper and made sense to me as to why this study and paper is important in a broader sense (the 'so what' question). The question of why this study is important to do and for whom (e.g. science or dance?) came up throughout still (though I do appreciate the several points of stated aims of the project and am not saying those do not answer to this question to an extant). Again I think its not something that can necessarily be solved with this paper, and is also part of the problem of doing an interdisciplinary project which impacts on your audience when sharing the work. Nevertheless it might be an issue to consider as you continue the work and distribute it further.

Referring back to the amount of information in the paper, it does seem there are so many variables to consider, including, as you note yourselves, personality traits. This point again made me question why the scientific paradigm was important then and what this desire is fuelled by.

Kinaesthesia and 'kinaesthetic togetherness' is stated as a key feature of the studies, your interests and the paper. It is interesting to note that I do not get a sense of 'kinaesthetic togetherness' from the paper overall, as yet. Your own admission for future research to 'collect first-person experiences of togetherness and aesthetic experience fromthe dancers themselves' (line 935) is encouraging on this last point. Kinaesthesia is a private experience and so very difficult to read from third person observation, as you have noted in parts of this paper. While I empathise with the limitations of word count and what you can report on and have summarised for us in the pilots, I agree that this area of the work can be represented better in future, as the participants' voices might make this issue of kinaesthetic togetherness come out more.

Author Response

Thank you again for very insightful comments.

- we have now clarified that although we correlate ratings of enjoyment and tempo difference, we see enjoyment as only one part of the aesthetic response (line 713). Also we added a clarifying sentence to this effect to discussion (line 784).

- we’ve rewritten the last paragraph, and now explicitly point out some possible future applications of our work, in a number of related fields, as well as some of our thinking about how to better capture the wider range of coordination patterns in improvisation, and decouple togetherness and synchronisation better.

- we’ve added a sentence on line 902 emphasising that methods of gathering and making use of the subjective experiences of improvisers is an important aspect we need to work on in the future studies.